# Association between Variants of the TRPV1 Gene and Body Composition in Sub-Saharan Africans

**DOI:** 10.3390/genes15060752

**Published:** 2024-06-07

**Authors:** Maddalena Giannì, Marco Antinucci, Stefania Bertoncini, Luca Taglioli, Cristina Giuliani, Donata Luiselli, Davide Risso, Elisabetta Marini, Gabriella Morini, Sergio Tofanelli

**Affiliations:** 1Dipartimento di Biologia, Università di Pisa, Via Ghini 13, 56126 Pisa, Italy; maddalena.gianni@univie.ac.at (M.G.); marco.antinucci@iit.it (M.A.); stef.bertoncini@gmail.com (S.B.); luca.taglioli@unipi.it (L.T.); daviderix@gmail.com (D.R.); 2Department of Evolutionary Anthropology, University of Vienna, 1030 Vienna, Austria; 3Central RNA Laboratory, Istituto Italiano di Tecnologia (IIT), 16163 Genova, Italy; 4Dipartimento di Scienze Biologiche, Geologiche e Ambientali (BiGeA), Università di Bologna, 40126 Bologna, Italy; cristina.giuliani2@unibo.it; 5Dipartimento di Beni Culturali (DBC), Università di Bologna, 48121 Ravenna, Italy; donata.luiselli@unibo.it; 6Dipartimento di Scienze della Vita e dell’Ambiente, Università di Cagliari, 09042 Cagliari, Italy; emarini@unica.it; 7Università di Scienze Gastronomiche, 62010 Pollenzo, Italy; g.morini@unisg.it

**Keywords:** TRPV1, sub-Saharan Africans, BIVA, body composition, adaptation

## Abstract

In humans, the transient receptor potential vanilloid 1 (*TRPV1*) gene is activated by exogenous (e.g., high temperatures, irritating compounds such as capsaicin) and endogenous (e.g., endocannabinoids, inflammatory factors, fatty acid metabolites, low pH) stimuli. It has been shown to be involved in several processes including nociception, thermosensation, and energy homeostasis. In this study, we investigated the association between *TRPV1* gene variants, sensory perception (to capsaicin and PROP), and body composition (BMI and bioimpedance variables) in human populations. By comparing sequences deposited in worldwide databases, we identified two haplotype blocks (herein referred to as H1 and H2) that show strong stabilizing selection signals (MAF approaching 0.50, Tajima’s D > +4.5) only in individuals with sub-Saharan African ancestry. We therefore studied the genetic variants of these two regions in 46 volunteers of sub-Saharan descent and 45 Italian volunteers (both sexes). Linear regression analyses showed significant associations between *TRPV1* diplotypes and body composition, but not with capsaicin perception. Specifically, in African women carrying the H1-b and H2-b haplotypes, a higher percentage of fat mass and lower extracellular fluid retention was observed, whereas no significant association was found in men. Our results suggest the possible action of sex-driven balancing selection at the non-coding sequences of the *TRPV1* gene, with adaptive effects on water balance and lipid deposition.

## 1. Introduction

TRP (Transient Receptor Potential) channels are evolutionarily conserved integral membrane proteins. They are structurally characterized by six transmembrane helices forming ion channels with variable cation selectivity [1], expressed in almost all tissues and cell types (GTEx Release V8).

In mammals, 28 TRP channels have been identified, grouped into subfamilies based on amino acid sequence homology: TRPC (‘canonical’), TRPM (‘melastatin’), TRPV (‘vanilloid’), TRPA (‘ankyrin’), TRPML (‘mucolipin’) and TRPP (or PKD) (‘polycystin’) [2]. These channels play essential roles in various physiological processes and participate in many sensory modalities [3].

They are involved in environmental sensing, being able to respond to a wide range of stimuli such as temperature, pH, osmolarity, pheromones, and plant compounds. In the oral cavity and nose, by transmitting signals generated by compounds in food to the brain, they give rise to the chemesthetic sensations of irritation, hotness, coolness, and pungency [4]. TRP channels play a key role in thermosensory perception and adaptation in several species [5,6].

TRPV (transient receptor potential vanilloid) channels were so named because the first identified member of this group, VR1, later named TRPV1, responded to the vanillylamide capsaicin, a chemesthetic compound produced by plants of the genus *Capsicum* [7]. Chemesthesis plays an important role in the body’s chemically activated defense mechanisms, the ‘chemofensor complex’ [8], used to avoid (or reduce) contact, inhalation, and ingestion of potentially harmful compounds. Interestingly, the TRPV1 receptor is also involved in innate immunity, being activated by N-acyl homoserine lactones, quorum-sensing molecules produced by Gram-negative bacteria [9]. This role is also demonstrated for taste receptors, in particular certain bitter receptors [10].

The *TRPV1* gene is expressed in 54 human tissues (GTEx Release V8). Activation of oro-pharyngeal TRPV1 following consumption of food containing hot compounds is perceived as high temperature. Physiological responses, such as gustatory sweating, are activated to counteract this input. The TRPV1 activation threshold temperature is lowered by vanilloids and many other natural compounds, such as piperine in pepper and gingerols in ginger. pH values below 6, a level easily reached by tissue injury such as infection and inflammation, also show this activity [11], giving TRPV1 an important role in the process of injury-related hyperalgesia, inflammation, and pain [12]. Activation of TRPV1 by vanilloids is followed by rapid and sustained desensitization [13,14], resulting in a particular form of analgesia, thus making TRPV1 a potential pharmacological target in pain therapy [15,16,17].

TRPV1 channels are also expressed in the central nervous system (CNS), where temperature and pH are strictly constant, suggesting the existence of endogenous brain agonists for this receptor, identified in endovanilloids such as anandamide and N-arachidonoyl-dopamine (NADA) [18,19,20].

The systemic response resulting from the activation of TRPV1 (among other TRPs) plays a role in adipocyte thermogenesis, adipogenesis, adipose tissue inflammation, and obesity [21,22,23], as well as water retention [24]. However, the relationship between TRPV1 variants and body composition in healthy individuals is a genotype-phenotype relationship that has not been explored. In previous papers PROP (6-n-propylthiouracil) phenotype (depending mainly from the bitter taste receptor gene *TAS2R38* genotype) has been proposed as indicator of body mass and adiposity, with contradictory results from different research groups [25,26,27,28].

The use of chili peppers is widely applied in several folk medicines to prevent and treat diabetes and other metabolic disorders [29,30,31], influencing, directly or indirectly, the energy balance and therefore body weight [32]. The mechanism of action of capsaicin in glucose control, energy homeostasis, and obesity-related diseases has been explained by both TRPV1-dependent and TRPV1-independent mechanisms [33].

Despite the involvement of TRPVs in important physiological and environmental signaling, little is known about their genetics, and very few papers have reported a systematic analysis of genomic data [34].

Previous studies have shown that the genetic variability at *TRPV1* includes splicing variants, sometimes with tissue-specific expression, with greater or lesser sensitivity to a specific category of stimuli (e.g., vanilloid agonists, [35]). Although variability in the sensory perception of hotness between and within populations is known, no study to date has been able to identify a relationship between this phenotypic trait and *TRPV1* variants [36]. Furthermore, the role of TRPV1 in systemic responses implies a relationship with other traits, such as pain perception [37] and inflammatory disease risk [38,39].

In this study, we addressed some fundamental questions about the role of TRPV1 channels in recent human evolution. How is the variability at the *TRPV1* gene distributed among human populations? Is this variability related to variations in environmental conditions? Is it related to a phenotype with adaptive relevance, like body composition?

To answer these questions, we first explored human variability upstream and downstream of the *TRPV1* gene through database searching and in silico analyses. Then, we collected both genotypic and phenotypic data from 46 volunteers of sub-Saharan origin and 45 volunteers of Italian origin to associate *TRPV1* variants with body composition and sensory perception.

## 2. Materials and Methods

### 2.1. Linkage Blocks Detection

A survey of genomes deposited at the International Genome Sample Resource (IGSR) and the Estonian Genome Centre Database (EGDP 483 high-coverage genomes, 148 populations worldwide) was performed. Within the IGRS, three datasets were considered: the 1000 Genome Project Phase 3 (KGP3, 2504 whole genomes, 26 populations worldwide), the Human Genome Diversity Project (HGDP, 828 individuals, 54 populations), and the Simons Genome Diversity Project (SGDP, 276 individuals, 129 populations). SNP frequencies were obtained from the ORF of the human *TRPV1* gene (transcript NM_080704.4) and a surrounding region of −5000 bp/+10,000 bp (GRCh38.p14 coordinate 17:3560446-3619411, Figure 1).

The analytical software PLINK version 1.9 [40] was used to calculate the minor allele frequency (MAF, --freq function) and genetic divergence between groups (Wright–Malécot Fst values, --fst function) for each SNP. Fst values above 0.15 were considered moderately high, and values over 0.25 were considered as high according to Frankham et al. [41]. PLINK v. 1.9 was also used to identify haplotype blocks in pairwise linkage disequilibrium (D’ and R^2^ > 0.75, --r2 and --ld-snp functions), and DNAsp [42] was used to detect natural selection signals (Tajima’s neutrality test).

### 2.2. Sample Composition

A group of 46 healthy sub-Saharan African donors (SSA) of both sexes was recruited and compared with an Italian (ITA) sample of 46 healthy volunteers (Table 1 and Appendix A). Exclusion criteria were: subjects with pulmonary, severe cardiovascular or uncontrolled metabolic diseases, electrolyte abnormalities, cancer, inflammatory conditions or using implanted electrical devices were excluded from the study. The mean age of the subjects was 32 ± 10.9 and 26 ± 4.9 years for the SSA and ITA groups, respectively.

The ancestry of the volunteers was verified by the residence on the identity document and by a self-declaration regarding the country of origin of the 2 parents.

By signing a consent form, volunteers agreed to complete a questionnaire on dietary habits, perform a sensory perception test, measure their body composition, and donate a saliva sample for genetic analysis.

### 2.3. Dietary Habits

A questionnaire was administered to reconstruct individual food consumption, as well as personal data and health conditions, including metabolic disorders (Appendix A). Food consumption frequency was assessed for seven hot foods (containing TRPV1 agonists) and three cooling foods (containing TRPM8 agonists). Questionnaire responses were organized into three categories: null (never), moderate (at least 2 times per month), and frequent (at least 2 times per week).

Also, questions were asked about hot, cold, and pain tolerance, categorized as low, medium, and high.

### 2.4. Perception Tests

A test was performed to assess the capsaicin perception threshold in each individual. Ten solutions of pure capsaicin were prepared by consecutive 1:2 dilutions in ethanol 99.3%, ranging from 2.243 μg/mL to 0.004 μg/mL. Then, 20 μL of each solution was aliquoted onto cotton swabs numbered 1 to 10, and ethanol allowed for drying out completely. Individuals were not informed of the substance they were about to taste. The swabs were tested sequentially, starting from the lowest concentration (swab 1), proceeding to the next higher concentration until the subject could feel a sensation. Volunteers were asked to hold each swab on the tongue for at least 5 s and to rinse their mouths with room temperature water before each new swab. The corresponding cotton swab number was recorded, and the intensity of the stimulus was indicated on a Labelled Magnitude Scale (LMS, [43]), with a scale of 0 to 100.

Since the correlation of PROP phenotype with body mass and adiposity were reported with divergent results from different groups [25,26,27,28], we decided to also perform a PROP perception test as reported in Risso et al. [44]. A 50 mM PROP solution (8.51 mg/mL) was prepared following the protocol with filter paper discs described in Zhao et al. [45]. At the end of the capsaicin test, a single swab of PROP was offered to be held in the mouth for 10 s. Subjects were asked to record taste intensity on an appropriate LMS.

Ten categories were identified for capsaicin perception threshold, corresponding to the lowest concentration perceived. Two categories were reported for PROP perception, following Drayna’s classification into “tasters” (score ≥ 50) and “non-tasters” (score < 50) phenotypes [46]. Each category was treated as a distinct phenotype in the genotype–phenotype and phenotype–phenotype correlation analysis (see below).

### 2.5. Anthropometry and Body Composition

The sampling phase took place in Italy during the temperate seasons, between late March and early June and in late September, to ensure that water loss levels were not affected by body acclimation. Individuals were asked to observe a complete fast of at least three hours (no food, no water) before the measurement.

Anthropometric measurements (height, weight, upper arm, waist, and circumferences) were taken according to standard international criteria [47]. Body mass index (BMI) was calculated as weight/height^2^ (kg/m^2^).

Bioelectrical values of resistance (R, ohm) and reactance (Xc, ohm) were measured through a portable Vitality AnalyzerTM bioimpedance device (IPGDX, LLC, Littleton, CO, USA), which applies a current of 0.6 microA at 50 kHz and using the standard positions for the outer and inner electrodes on the right hand and foot.

To evaluate body composition, specific Bioelectrical Impedance Vector Analysis (specific BIVA; [48]) was applied. BIVA allows the graphical and statistical analysis of bioelectrical vectors, as defined by their module, that is, impedivity (Z = (R^2^ + Xc^2^)^0.5^, ohm), and their inclination, that is, phase angle, (PhA = arctangent Xc/R * 180/π, degrees).

Following the specific BIVA approach, R and Xc were standardized by a correction factor A/L, where A represents an estimate of the transverse area of the body (0.45 arm area + 0.10 waist area + 0.45 calf area) and L the distance between electrodes (height*1.1) [48]. According to Ohm’s law, this correction reduces the influence of body size and shape on bioelectrical variables (specific resistance, Rsp, ohm*cm; specific reactance, Xcsp, ohm*cm), which are therefore mainly related to body composition variability (data are shown in the Appendix A).

As in the classic vectorial approach [49], two types of graphs represent the output of specific BIVA on the plane defined by specific resistance on the *x*-axis and specific reactance on the *y*-axis: the tolerance ellipses and the confidence ellipses. Tolerance ellipses represent the bioelectrical variability of the reference population; individual or mean vectors can be plotted on the graph, and their body composition can be evaluated depending on their position. According to specific BIVA [48], the major axis of specific tolerance ellipses, mainly due to variations in vector length and Rsp, is related to variations in relative fat mass content (FM%), with higher values towards the upper pole. The minor axis, mainly due to variations in PhA and Xcsp, is related to body cell mass (higher values on the left) and extracellular-to-intracellular water ratio (ECW/ICW) (higher values on the right) and is considered a proxy for muscle mass and quality. Confidence ellipses represent the 95 percent confidence interval around the sample mean and allow the graphical comparison among samples, with no overlapping ellipses indicating significant differences [49]. The statistical difference between confidence ellipses can be evaluated by the Hotelling T^2^ test [50].

Samples were grouped by sex, as muscle and fat mass and distribution are significantly different between males and females [51], with the latter showing longer impedance vectors and lower phase angle [48]. When focusing on a phenotypic trait, regardless of the sex, we considered males and females together after standardization (Z scores).

To evaluate sample distribution patterns and identify outliers, the vectors of each group were plotted on the specific tolerance ellipse representing the best approximation of the source population. ITA vectors were plotted on an Italo-Spanish reference sample (213 males, 227 females, aged 18–30 years, [52]). SSA vectors were plotted on an African American reference sample (181 males, 175 females, aged 18–49 years), representing a subsample of the NHANES dataset analyzed by Buffa et al. [48], as body composition reference data from the African continent are still lacking in the literature.

### 2.6. DNA Sampling and Genotyping

A volume of 2 mL of saliva was collected with the Oragene™ DNA Self-Collection Kit (DNA Genotek, Ottawa, ON, Canada) and stored at room temperature for several weeks before extraction. Whole DNA extraction was performed with the prepIT©L2P Laboratory (DNA Genotek Inc., Canada) protocol according to the manufacturer’s instructions.

Two SNP-rich segments identified in the *TRPV1* gene region (herein referred to as R1 and R2, Figure 1) were selected for PCR amplification and Sanger sequencing. To ensure the absence of polymorphisms within the primer sequences, the region was screened to identify all SNPs, using the USCS (genome.ucsc.edu) and ENSEMBL (ensemble.org) genome browsers. Candidate primers were generated with Primer3Plus software (v. 3.3.0) [53]. Melting temperature and absence of dimerization were assessed with AutoDimer (implemented in STRBase 2.0, [54]). Finally, the ability of primers to match sequences of other species was tested with NCBI BLAST [55]. The total amplicon length was set to 699 bp and 508 bp for R1 and R2 segments, respectively, taking care to allow a distance of 20–30 bp from the outermost polymorphic locus (Appendix A).

DNA was sequenced by the Cycle Sequencing method using the BigDye™ Direct Cycle Sequencing Kit (ThermoFisher Scientific, Waltham, MA, USA) and an ABI 3730xl DNA Analyzer system (phred: 20–1100 bp).

Genotype calling was performed manually by aligning FASTA files to the GenBank reference sequence (GRCh38) using BioEdit 7.7 software [56]. For R2 genotype calling, the forward strand was considered, whereas R1 required the sequencing of both strands, forward and reverse, due to the presence of a tetranucleotide STR (nsv1874533, 17:3596296-3596321).

### 2.7. Data Analysis

Subjects were divided into groups according to ancestry (ITA and SSA) and sex. SSA was subsequently grouped according to the diplotype. Questionnaire responses organized into frequency or intensity categories were compared to capsaicin perception to assess the environmental component of diet on taste perception. Capsaicin perception threshold distribution among groups was also tested for association to sex, ancestry, and diplotype (for SSA only).

The specific BIVA approach was used to assess body composition. To assess the difference between subgroups, the Hotelling T^2^ test was applied to confidence ellipses, considering the center of each ellipse as the bivariate mean of Rsp and Xcsp, and setting the significance threshold at *p* = 0.05. The Mahalanobis distance index (D^2^, [57]) and Fisher F indices were also used to estimate the distance between the two data distributions. When calculating D^2^, the critical values of variables Rsp and Xcsp correspond to the perimeter of the 95% confidence ellipses.

The representativeness of the sample was tested by comparing allele frequencies and haplotype occurrence with the KGP3 group of Tuscan individuals (TSI) and the AFR(-ASW) group (all African subgroups except ASW, which include individuals with African American ancestry). Minor allele frequency (MAF, --freq) and linkage disequilibrium (LD, --r2 and --ld-snp) at each locus were calculated using PLINK v. 1.9 software [40].

For SSA only, the correlation between each diplotype–phenotype pair was assessed by performing ANOVA and linear regression tests (--linear --covar). Ten phenotypes were distinguished for capsaicin perception, corresponding to the 10 perception thresholds. In addition, the PROP perception intensity distribution was calculated and correlated to BMI values.

To test the correlation between genetic variability and body composition, the four bioelectric variables (Rsp, Xcsp, Zsp, PhA) were considered as four phenotypes and tested independently against each diplotype. To avoid sex bias on body composition, sex was considered as a covariant in the regression model.

To investigate the effects of climate differences, SSA was divided into West Africans (22 subjects) and East Africans (24 subjects), and association was measured with haplotype status.

## 3. Results

### 3.1. In Silico Analysis

A total of 87 SNPs showing Fst > 0.25 (*p* < 0.001) were identified in the ORF of the *TRPV1* gene and surrounding regions (Figure 1, Appendix A).

**Figure 1 genes-15-00752-f001:**
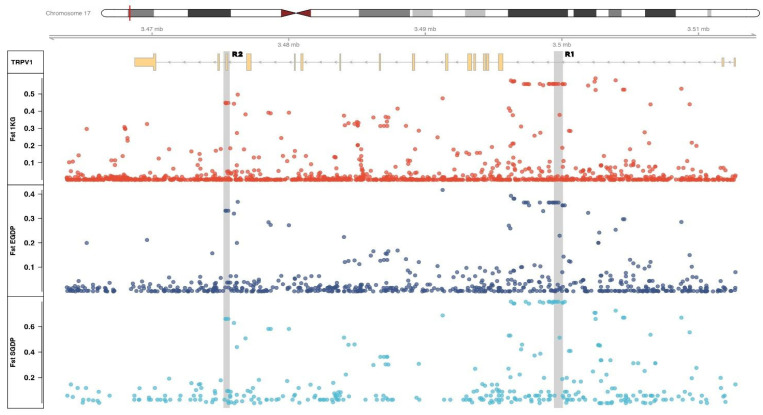
Fst values for SNPs within and around the *TRPV1* gene region (17:3560446-3619411) obtained from the 1 KGP Phase 3 dataset (red), EGDP dataset (dark blue), and SGDP (clear blue). Genotyped regions R1 and R2 are highlighted.

The highest Fst values were found for 2 sets of SNPs. The first one (R1) accounts for a region over 13 k bp long, from position 17:3592966 to 17:3606044, and counts 31 loci (Fst GKP3 = 0.59, EGDP = 0.37, SGDP = 0.65) (Table 2). This region overlaps an enhancer region (ENSR00000547483), an open chromatin region (ENSR00000547484), and one CTCF binding site (ENSR00000547485). The second set (R2) spans a region of about 900 bp, from position 17:3572073 to 17:3572970, and includes 5 loci (Fst KGP3 = 0.45, EGDP = 0.33, SGDP = 0.54) (Table 2).

LD tests revealed that only in sub-Saharan African populations do these SNPs cluster into two linkage blocks. In this study, for convenience, we will call H1 the 31-SNP haplotype that includes the R1 segment and H2 the 5-SNP haplotype that includes the R2 segment. In sub-Saharan African populations, alternative alleles showed frequencies about 70% in H1 loci and 65% in H2 loci. These same alleles are rare (0–1%) in non-African groups.

Tajima’s neutrality tests calculated on the IGSR (KGP3) yielded highly positive values, D = +4.51 (*p* < 0.001) and D = +3.36 (*p* < 0.01) for R1 and R2, respectively. D calculation for other datasets was impossible due to the low number of African individuals. Tajima’s values support the hypothesis that the *TRPV1* region underwent balancing selection or rapid population contraction in the African continent (Appendix A).

### 3.2. Haplotype Characterization

Segments R1 and R2 were successfully sequenced, and genotypes were correctly assigned to all recruited subjects. Segment R1 (699 bp, 17:3596108-3596806), located within the 5′UTR, scored 8 polymorphisms. The R2 segment (508 bp, 17:3571903-3572410), located between introns 14–15 and 15–16, scored a total of 6 polymorphic loci (Table 3).

Five of the eight SNPs of R1 are included in the H1 haplotype, while three of the six SNPs of R2 are included in H2. All genotyped SNPs that are not included in the two blocks showed Fst values lower than 0.15, except for one (rs161383, Fst = 0.19). According to the KGP3 dataset, rs161383 is in linkage with only one locus, located upstream the H1 block. For all these SNPs, the observed MAFs corresponded to the European and African frequencies of the IGSR (Table 3 and Figure 2). In R1, reference allele frequencies ranged from 43 to 44% in the SSA sample, compared with 36% in AFR(-ASW) of the KGP3 dataset, while they were 0% in the ITA sample, in good agreement with the 0.5% of TSI. Similarly, reference allele frequencies in R2 ranged from 41 to 49% in SSA versus 50% in AFR(-ASW), and they were 0% in ITA and TSI.

When the SNPs in the SSA sample were tested for LD, they clustered according to the linkage blocks detected in the IGSR and EGDP databases (Table 3). The exonic locus rs877610 was polymorphic in the sample, and its MAF (SSA = 0.12%. ITA = 0.04%) was similar to the value of KGP3 (AFR(-ASW) = 0.13%, TSI = 0.03%). This locus maps within the H2 region but is not in linkage with it. Its derived allele produces the synonymous mutation K719K.

Because of the low number of sampled SNPs, genotypes could not be reliably phased. However, because the haplotype states reflect well the allele frequencies of larger datasets, we assumed that they have the same phase. Therefore, we characterized the diplotype (the genotype of a haplotype block) of each individual in our sample. We identified three haplotype states for H1 and two for H2 (Table 4).

### 3.3. Perception Patterns and Correlation to Diplotypes

The frequency of capsaicin perception thresholds (swabs 1 to 10) rapidly dropped after swab 5 (0.07 ug/mL) and became zero at swab 9 (1.12 ug/mL) (Appendix A).

SSA and ITA had similar trends, and the differences can be considered as random fluctuations (Student’s *t* test, *p* = 0.74) (Appendix A for this and following results). The distribution of perception thresholds was not significantly different by sex (Student’s *t* test, *p* = 0.16). However, males showed a slightly higher threshold (Mean = 3.9, Mode = 5) than females (Mean = 3.4, Mode = 2). No significant relationship was found between H1 or H2 diplotypes and the chemesthetic response to capsaicin (*p* = 0.63 for H1, *p* = 0.72 for H2).

Capsaicin perception was also tested for association with frequency of food consumption. The Pearson correlation index was always lower than 0.40, the highest correlation being the one with chili pepper in the SSA group (r = 0.39) (Appendix A).

Fewer individuals were classified as PROP non-tasters (21.7% of Italians, 22.2% of sub-Saharans have intensity < 50) compared to tasters. PROP perception phenotypes and BMI were found to be not significantly correlated (Appendix A).

### 3.4. Body Composition Patterns and Correlation to Diplotypes

When each group was plotted on the chosen reference, the mean values were found to be within the 50% tolerance ellipse, indicating that each reference population well approximates the source population of each sampled group (Appendix A). Individual bioelectrical vectors were also within the range of the reference population, with only one outlier in the SSA women group (SSA_F, Appendix A).

As expected, males and females showed distinct impedance distributions (T^2^ = 8.7, *p* = 0.016, see Table 5) regardless of ancestry, with women showing higher Zsp values and narrower PhA than men (Appendix A).

Sub-Saharan Africans and Italians showed significantly different distributions when standardized for sex (T^2^ = 13.5, *p* = 0.002, see Figure 3a). The differences based on ancestry were driven by the female component. Specifically, the confidence ellipses of ITA and SSA males overlapped widely (ITA_M vs. SSA_M: T^2^ = 1.5, *p* = 0.49), while the ellipses of females did not (ITA_F vs. SSAF: T^2^ = 21.9, *p* = 0.0003) (see Figure 3b).

SSA and ITA women differed mainly in vector length (Zsp = 502.10 ± 106.44 for SSA_F and Zsp = 407.38 ± 70.44 for ITA_F). The phase angle also differs, with lower values in ITA women (PhA = 5.67 ± 0.60) than in SSA women (PhA = 6.18 ± 0.62) (*p* = 0.02, see Appendix A). Although the presence of one African outlier partially drives this difference, removing one individual negatively affects the statistical power of the test, due to SSA_F’s small sample size.

When SSA was divided according to geographic origin (East or West Africa), body composition patterns were not significantly different (T^2^ = 2.0, *p* = 0.379), and the confidence ellipses overlapped widely (Appendix A).

SSA_F and SSA_M were then grouped by diplotype (H1-aa/ab/bb and H2-aa/ab/bb). The Hotelling T^2^ test showed significant bioelectrical differences between diplotype H2-aa and the two diplotypes H2-bb and H2-ab, when using the total sample with values standardized by sex (Table 5). To better understand the sexual contribution to this diversity, men and women were also analyzed separately (Figure 4).

In women, comparison between single diplotypes was not possible, due to the small sample size. Although, when heterozygous women H2-ab were grouped with homozygous H2-bb, a significant difference was observed with respect to H2-aa women (T^2^ = 23.8, *p* = 0.002) (Table 5). This difference is reflected in the trend shown on the tolerance ellipse (Figure 4a), where H2-aa tends to have a shorter vector Zsp and a larger PhA than the other two diplotypes.

Men did not show significant differences according to diplotype. Also, the tolerance ellipse in Figure 4b shows no trend in BIVA patterns. No significant difference was noted even when merging H2-ab and H2-bb in the same group.

The same tests were performed on H1 diplotype groups, but no significant difference was detected. Haplotype H1-c was found in two individuals only and was thus grouped with H1-b, as all variants except one have the same allelic state in both haplotypes. Still, a trend is visible in the women tolerance ellipse (Figure 4a).

Linear regression, with sex as the covariate, showed a significant association between diplotypes and BIVA variables Zsp, PhA, and Rsp (*p* < 0.05 in all cases, Table 6). The driving factor of body composition differences among SSA was further investigated by grouping per sex and ancestry (Table 7). In agreement with the Hotelling test reported in Table 5, group SSA_F showed highly significant differences in BIVA values in association to H2 diplotypes.

## 4. Discussion

Despite the potential involvement of the *TRPV1* gene in adaptive processes, associations between its variants and phenotypes have not been studied across human populations to date. In this research, two highly polymorphic LD blocks (named H1 and H2) were identified in silico in the ORF and the 5′UTR upstream region of the *TRPV1* gene, showing patterns compatible with a process of balanced selection. Their evolutionary significance was explored by measuring the association between diplotype states, body composition, sensory perception, and dietary habits.

At the level of blocks H1 and H2, we identified haplotypes showing high MAF (43–49%) and highly positive Tajima’s D values (D > +4.5) only in the sample with sub-Saharan ancestry. A positive D reflects episodes of population subdivision/contraction or balanced selection [58], and it is not trivial to discern between the two alternatives. In this case, however, the influence of demography can reasonably be considered negligible, as the excess of pairwise mutation differences over the number of segregating sites has been observed on a continental scale and localized to a short nucleotide sequence, with no impact on the genome as a whole.

The fact that, both in the databases and in the sampled groups, no major deviations from Hardy–Weinberg expectations were observed can be interpreted as the effect of natural selection operating over long periods of time. Presumably, the *TRPV1* region has escaped previous genomic scans [59,60,61,62,63] due to the small size of the regions involved (2.7% of the total gene length), the difficulty in simulating different types of stabilizing selection, and the low power of statistical approaches, which ignore intragenic recombination [64].

Our results suggest that the selective agent is to be sought in a different history of human–environment interaction in Africans and non-Africans.

Dietary habit was the candidate agent we analyzed first. The TRPV1 agonist capsaicin is involved in weight loss by decreasing appetite and by increasing fat mobilization and insulin/leptin resistance [33]. The process is coupled to an increase in brown adipose cells and a decrease in white adipogenesis [65]. Nonetheless, no significant association of diplotypic states was observed with sensitivity to capsaicin (or to PROP), BMI, or food consumption. Genetic influence may be masked by polygeny or by exogenous factors such as frequency of consumption or the synergistic action exerted by the gut microbiota ([36], Vinerbi et al. in preparation). Further research on this regard is in progress.

Comparative studies examining animal species adapted to different thermal environments demonstrated how changes in TRPV1 heat responses (but not to capsaicin or acids, thereby maintaining its function as a detector of chemical cues) arise from just one amino acid difference in the orthologous genes [66]. Therefore, physiological response to climatic conditions, which is known to strongly influence body composition and shape, was the second candidate agent analyzed, and it has to be intended here as the complex of processes that, in the long term, guarantee osmotic and energetic homeostasis. As we have observed experimentally through anthropometric and impedance measurements, the SSA and ITA groups are clearly distinct in terms of body composition. Sub-Saharan Africans recorded higher Zsp and PhA values, indicative of a higher percentage of fat mass (FM%) and a lower extracellular/intracellular water ratio (ECW/ICW). This could be due to a different lifestyle, but it is also consistent with an evolutionary adaptation to long periods of drought and famine. Indeed, in an arid environment, heat loss and transpiration are maximized by increased surface area/volume ratio and fat accumulation [67]. Fat deposition/mobilization and water retention are controlled by sexual hormones, so their regulation is different in the two sexes [68,69]. This explains why the sampled women showed higher values of FM% and ECW/ICW, regardless of ancestry. Furthermore, SSA women show higher FM% and lower ECW/ICW than ITA women. Interestingly, we found that in SSA individuals, the BIVA values were correlated with *TRPV1* H1-b and H2-b haplotypes, in both homozygous and heterozygous states. These haplotypes are absent in non-African populations as the relevant variants are almost monomorphic and not in linkage.

Regarding the ECW/ICW ratio, it is known that a state of hyperosmolarity of the circulating blood corresponds to an activation of TRPV1 in the central nervous system, which leads to an increase in water retention at the systemic level [24]. During episodes of hyperthermia, when the sympathetic nervous system (SNS) and the hypothalamic–pituitary–adrenal (HPA) axis trigger processes that favor water loss (sweating, tachypnoea, skin vasodilation, salivation), TRPV1 induces the same compensatory reactions [70,71].

Regarding FM%, it is known that TRPV1 and lipids mutually interact to regulate their expression [22,65,72]). Various types of lipids, such as phospholipids, triglycerides, and steroids (including estrogen and oxytocin), influence the gating activity and/or expression of *TRPV1* [73,74,75,76,77]. On the other hand, TRPV1 channels influence lipid metabolism in a complex manner that has so far yielded contradictory results. Decreased *TRPV1* expression in mice has been found to protect against diet-induced obesity [78] and promotes oxygen consumption, fat oxidation, and locomotor activity [79]. In contrast, loss of the *TRPV1* gene in Western-fed mice causes hyperlipidemia, and animals exhibit reduced locomotor activity with a more pronounced effect in males than in females [80,81]. TRPV1 agonists, such as vanilloids, capsaicin, and oxytocin, influence lipid metabolism, mainly by reducing lipid deposition. By activating TRPV1 channels, they cause an increase in intracellular free Ca^2+^ levels, thus triggering the desensitization of nociceptive neurons [82] and the suppression of visceral fat accumulation through the upregulation of UCP1 (Mitochondrial Uncoupling Protein 1) [83]. Lipid metabolism has also been found to be regulated by the interaction of *TRPV1* with transcription factors involved in energy homeostasis, such as peroxisome proliferation-activated receptors (PPARs, [84,85]) and sterol-responsive element-binding proteins (SREBPs, [86,87]).

Our results suggest a role for the H1 and H2 haplotypes as sequences harboring binding sites that regulate the activation/suppression of *TRPV1* expression by different mechanisms, acting independently or synergistically to enhance fat storage efficiency. They have not yet been associated with an altered expression level of *TRPV1*, and no SNPs of the two blocks are present in the GTEx or RegulomeDB browsers. However, the overlap of H1 with an enhancer, an open chromatin region, and a transcription repressor (CTCF) suggests that these mechanisms are involved in the regulation of TRPV1 channel expression in peripheral tissues. Furthermore, the overlap with *TRPV3* and *SHK* gene regions may suggest that H1 and H2 play a regulatory role in these genes in addition to or instead of *TRPV1*.

Balanced SNP/haplotype frequencies and neutrality tests suggest that the most likely cause of the association between *TRPV1* haplotypes and body composition is balancing selection. A prerequisite for this selective process is the greater fitness of heterozygous than homozygous diplotypes or the frequency-dependent fluctuation of alleles around an average value. The frequency pattern of *TRPV1* alleles in sub-Saharan populations does not faithfully obey the model of balancing selection based on heterozygous advantage. The framework found is more compatible with a sex-specific directional selection, based on the higher fertility (fitness) of females, who more efficiently accumulate subcutaneous fat, and with an enhancing effect of sexual selection, which promotes the male selection of females with pronounced gynoid forms (low waist-to-hip ratio), according to local beauty standards [88] (Yu & Shepard, 1998).

Women show a greater capacity than men to store excess free fatty acids, obtained during periods of energy surplus, in the form of subcutaneous adipose tissue (SAT) [69,89]. This excess fat mass is highly mobile, and lipids can be easily recovered to get through pregnancy and lactation under conditions of prolonged nutrient deprivation. The SAT is rich in brown fat, which is highly supplied by blood vessels and thus serves as an easily exploitable water reserve. Furthermore, unlike visceral adipose tissue (VAT), SAT has low lipase activity and protects against diet-dependent ectopic fat formation, thus providing protection against cardiovascular disease and type 2 diabetes [90,91,92].

An extreme case of this phenomenon is the steatopygia of African hunter-gatherers, such as the Khoi-khoi of southern Africa or the pygmy groups of West Africa [93]. A sex-dependent role of *TRPV1* in SAT metabolism is also suggested by the fact that estrogen regulates *TRPV1* in the endometrium of immature rats [94] and in the arterial and bladder smooth muscle of post-pubertal female rats [95].

## 5. Conclusions

For over 200 thousand years, our ancestors lived in African environments under physical, hydric, and thermal stress. Hence, a more efficient defense strategy to maintain energy and osmotic homeostasis should be subjected to strong selective pressure. *TRPV1* (and other TRP family genes) are certainly important signaling molecules critically involved in the systemic response that regulate energy and water fluxes during heat defense, cold defense, thermally or mechanically induced pain, and acute inflammation. Thus, increased efficiency in fat and water balance can be interpreted as the effect of an adaptive response to the environment.

With the limitations due to small sample size, heterogeneity in the analyzed samples, uncontrolled confounding variables, such as physical activity levels, and narrow, sparse geographic coverage, the results of the present study propose that *TRPV1* H1 and H2 *a/b* haplotypic variants play a role in estrogen-mediated lipid deposition and fluid retention as an adaptive response to prolonged resource depletion and/or extreme thermal fluctuations. Further research efforts in larger sample sizes with higher statistical power are needed to unravel the molecular, cellular, and evolutionary mechanisms that led to the balancing of their frequencies in sub-Saharan Africans and the fixation of H1/H2 *a* variants in non-Africans.

## Figures and Tables

**Figure 2 genes-15-00752-f002:**
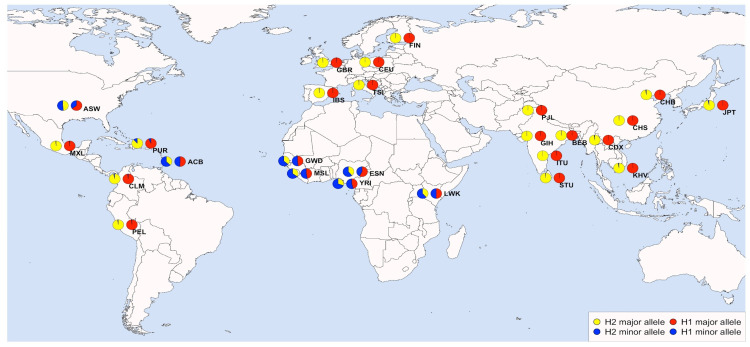
Allele frequencies for variant rs10491215 (representative of haplotype H1) and rs224548 (representative of haplotype H2) for the 26 populations of 1 KGP Phase 3. Pie chart distribution approximately represents populations’ location.

**Figure 3 genes-15-00752-f003:**
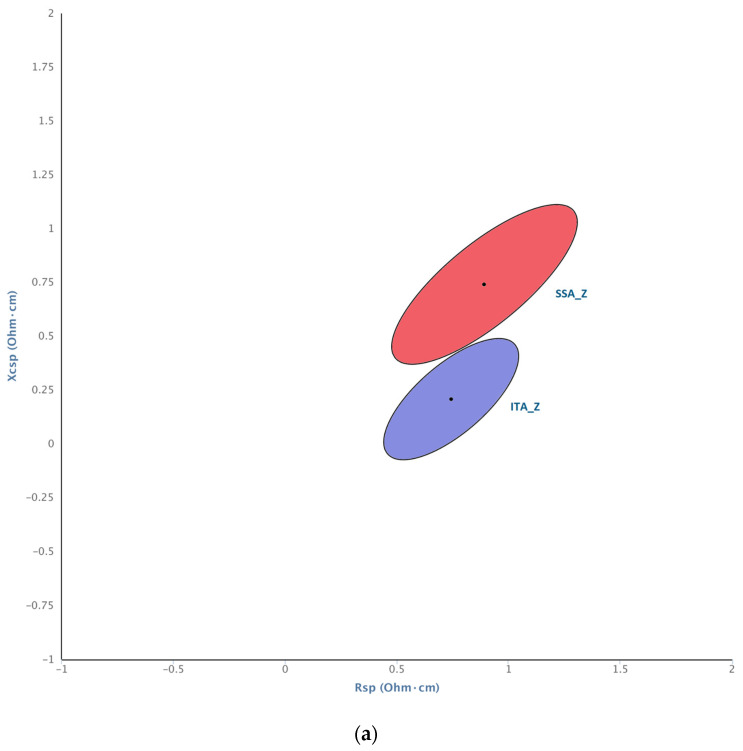
BIVA confidence ellipses showing population mean (vector) and 0.95 confidence interval (ellipse perimeter) for African and Italian samples. (**a**) Without sex distinction: African sample (SSA_Z, red) and Italian sample (ITA_Z, blue); (**b**) with sex distinction: African males (SSA_M, red), African females (SSA_F, pink), Italian males (ITA_M, dark blue), Italian females (ITA_F, clear blue).

**Figure 4 genes-15-00752-f004:**
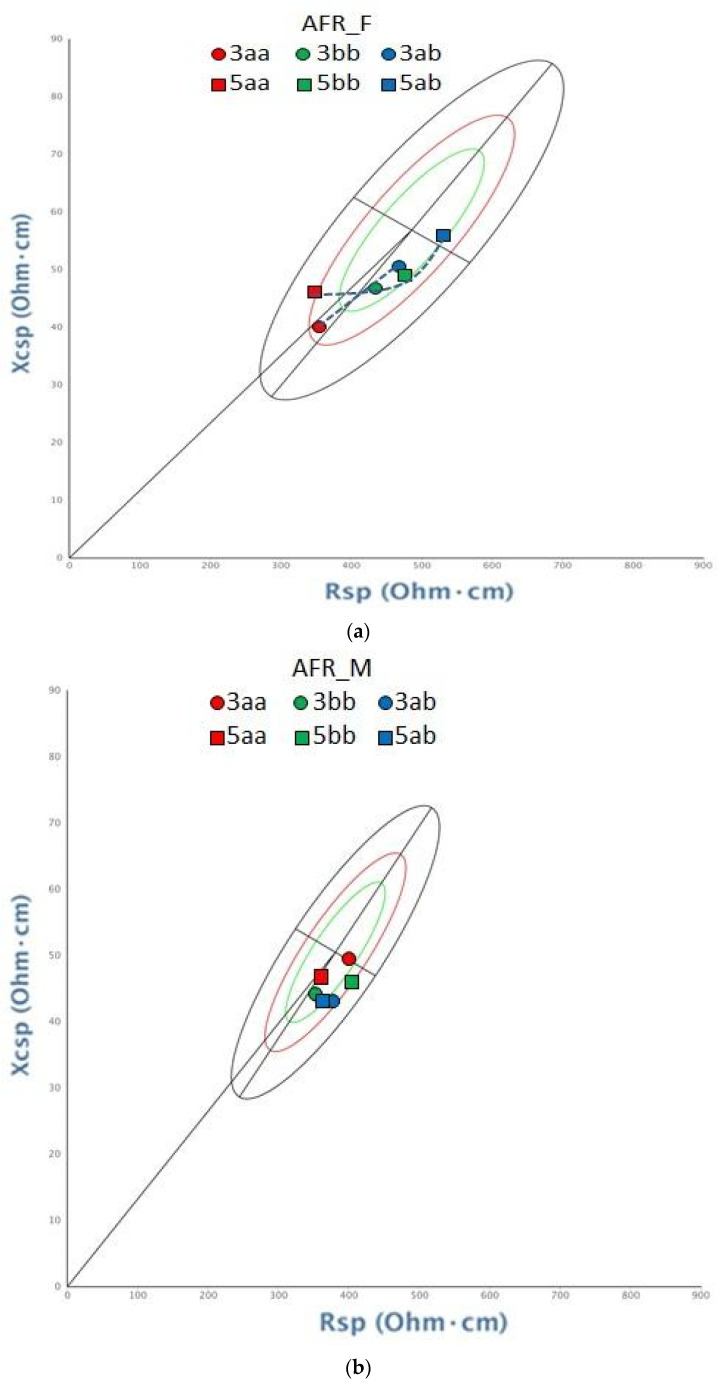
BIVA tolerance ellipses showing vectors for (**a**) African female sample and (**b**) African male sample, grouped by diplotype for the two haplotypic blocks. A population of African American women and men, respectively, is taken as a reference.

**Table 1 genes-15-00752-t001:** Sample composition: sample size and age (mean ± SD) for sub-Saharan African and Italian samples.

	Sub-Saharan African (SSA)		Italian (ITA)	
	N	Age	N	Age
males	34	33 ± 12	21	26 ± 6
females	12	29 ± 9	25	26 ± 3
total	46	32 ± 11	46	26 ± 5

**Table 2 genes-15-00752-t002:** Features of R1 and R2 *TRPV1* gene regions.: minor allele frequencies (MAFs) found at segments R1 and R2 in the SSA and ITA samples are consistent with 1 KGP phase 3 data.

Segment ID	Polymorphic SNPs	Block Location * (bp)	Regulatory Features	Category	Regulatory Feature Location * (bp)
R1	31	17:3592966-3606044	ENSR00000547483	Enhancer	17:3594201-3595323
			ENSR00000547484	Open chromatin	17:3596858-3597480
			ENSR00000547485	CTCF binding site	17:3597201-3597400
R2	5	17:3572073-3572970	N/A	N/A	N/A

* Genomic positions refer to GRCh38.p14 assembly.

**Table 3 genes-15-00752-t003:** Linkage blocks at R1 and R2 *TRPV1* gene regions.

Segment	SNP ID	Position * (bp)	LD Block	MAF KGP3 Total	MAF TSI	MAF ITA	AFR (-ASW)	MAF SSA
R1	rs10491215	17:3596159	H1	C = 0.178	C = 0.005	C = 0.00	G = 0.359	G = 0.433
R1	rs73303323	17:3596281	H1	C = 0.178	C = 0.005	C = 0.00	T = 0.359	T = 0.444
R1	rs73303325	17:3596443	H1	T = 0.178	T = 0.005	T = 0.00	C = 0.359	C = 0.442
R1	rs73303327	17:3596528	-	A = 0.128	A = 0.014	A = 0.033	A = 0.453	A = 0.378
R1	rs161383	17:3596724	-	G = 0.229	G = 0.224	G = 0.189	G = 0.038	G = 0.044
R1	rs7211511	17:3596744	H1	T = 0.178	T = 0.005	T = 0.00	C = 0.360	C = 0.433
R1	rs7211517	17:3596752	H1	G = 0.178	T = 0.005	T = 0.00	C = 0.360	C = 0.433
R1	rs114890125	17:3596764	-	T = 0.025	T = 0.000	T = 0.00	T = 0.094	T = 0.043
R2	rs224548	17:3572073	H2	G = 0.866	T = 0.000	T = 0.00	G = 0.497	T = 0.444
R2	rs224549	17:3572111	H2	T = 0.868	C = 0.000	C = 0.00	T = 0.498	C = 0.411
R2	rs877610	17:3572196	-	T = 0.070	T = 0.028	T = 0.011	T = 0.153	T = 0.122
R2	rs224550	17:3572270	H2	T = 0.866	C = 0.000	C = 0.00	T = 0.498	C = 0.489
R2	rs877611	17:3572271	-	C = 0.244	C = 0.299	C = 0.322	C = 0.178	C = 0.122
R2	rs877612	17:3572292	-	T = 0.070	T = 0.028	T = 0.011	T = 0.153	T = 0.122

KGP3 = Thousand Genome Phase 3 Dataset, TSI = KGP3 Tuscans (Italy), ITA = Italian sample, AFR(-ASW) = KGP3 Africans but ASW, SSA = sub-Saharan African sample. * Genomic positions refer to GRCh38.p14 assembly.

**Table 4 genes-15-00752-t004:** Fst values and haplotype states of R1 and R2 *TRPV1* allele variants in the genotyped samples: chromosome and gene locations according to GRCh38 assembly.

SNP ID	Position * (bp)	Gene Region	KGP3 Fst	SGDP Fst	EGDP Fst	H1-a	H1-b	H1-c
R1								
rs10491215	17:3596159	intronic	0.558	0.652	0.365	G	C	C
rs73303323	17:3596281	intronic	0.558	0.652	0.365	T	C	C
rs73303325	17:3596443	intronic	0.558	0.652	0.365	C	T	G
rs7211511	17:3596744	intronic	0.557	0.647	0.354	C	T	T
rs7211517	17:3596752	intronic	0.557	0.647	0.354	C	G	G
R2								
rs224548	17:3572073	intronic	0.448	0.535	0.332	G	T	
rs224549	17:3572111	intronic	0.445	0.535	0.332	T	C	
rs224550	17:3572270	intronic	0.448	0.535	0.332	T	C	

* Genomic positions refer to GRCh38.p14 assembly, positive strand.

**Table 5 genes-15-00752-t005:** Comparison of bioelectrical mean vectors between sexes, populations, and H1/H2 *TRPV1* haplotypes.

Groups	Used n	Heal n	T^2^	F	*p*	D
ITA vs. SSA	91	91	13.5	6.7	**0.002**	0.77
M vs. F	91	91	8.7	4.3	**0.016**	0.63
ITA_M vs. SSA_M	54	54	1.5	0.7	0.492	0.34
ITA_F vs. SSA_F	37	37	21.9	10.6	**0.0003**	1.65
SSA-W vs. SSA-E	46	46	2.0	1.0	0.379	0.42
Z-H2aa vs. Z-H2bb	19	19	8.3	3.9	**0.040**	1.32
Z-H2aa vs. Z-H2ab	36	36	68.8	33.4	**0.000**	3.19
Z-H2bb vs. Z-H2ab	37	37	5.4	2.6	0.087	0.86
Z-H1aa vs. Z-H1bb	25	25	3.3	1.6	0.229	0.71
Z-H1aa vs. Z-H1ab	32	32	1.2	0.6	0.555	0.40
Z-H1bb vs. Z-H1ab	33	33	1.3	0.6	0.534	0.40
M-H2aa vs. M-H2bb	17	17	5.6	2.6	0.107	1.15
M-H2aa vs. M-H2ab	25	25	2.7	1.3	0.300	0.70
M-H2bb vs. M-H2ab	26	26	3.2	1.5	0.238	0.74
M-H1aa vs. M-H1bb	19	19	3.5	1.7	0.221	0.87
M-H1aa vs. M-H1ab	26	26	4.5	2.1	0.143	0.93
M-H1bb vs. M-H1ab	23	23	1.9	0.9	0.424	0.55
F-H2aa vs. F-H2bb	*6*	2	N/A	N/A	N/A	N/A
F-H2aa vs. F-H2ab	*13*	11	N/A	N/A	N/A	N/A
F-H2bb vs. F-H2ab	*13*	11	N/A	N/A	N/A	N/A
F-H1aa vs. F-H1bb	*7*	5	0.8	0.3	0.733	0.68
F-H1aa vs. F-H1ab	*9*	9	3.5	1.5	0.288	1.25
F-H1bb vs. F-H1ab	*8*	6	0.9	0.4	0.691	0.69
M-H2aa vs. M-H2bb+ab	34	34	3.5	1.7	0.199	0.76
M-H1aa vs. M-H1bb+ab	34	34	3	1.5	0.247	0.70
F-H2aa vs. F-H2bb+ab	12	12	*23.8*	*10.9*	* **0.002** *	*3.18*
F-H1aa vs. F-H1bb+ab	12	12	1.3	0.6	0.591	0.71

M = males; F = females; Z = standardized values for both M and F; n = sample size, given by the sum of sample size of each element of the pair; T^2^ = Hotelling test; F = Fisher exact test; *p* = *p* value of T^2^ and F; D = Mahalanobis distance. Pairwise comparison: First section: African individuals grouped by sex and diplotype. In Italics: used sample size is different from real sample size. Second section: African individuals grouped by haplotype only. Third section: African individuals grouped by sex and presence/absence of haplotypes -H2b and -H1b, respectively. In Italics: sample was duplicated to obtain R and Xc standard deviation values. N/A: sample too small for statistical test. Probability valued below 0.05 are bolded.

**Table 6 genes-15-00752-t006:** Association between H1/H2 *TRPV1* haplotypes, BIVA variables, and capsaicin perception with sex as covariant.

Haplotype	Parameter	β	T Test	*p*
H1	Capsaicin perception	0.71	0.98	0.3354
H1	Rsp	−105.60	−4.88	**0.0170**
H1	Xcsp	−3.90	−1.49	0.1430
H1	Zsp	−105.50	−4.86	**0.0180**
H1	PhA	1.28	3.66	**0.0007**
H2	Capsaicin perception	0.69	0.94	0.3507
H2	Rsp	−102.90	−4964.00	**0.0130**
H2	Xcsp	−4234.00	−1609.00	0.1160
H2	Zsp	−102.90	−4.94	**0.0140**
H2	PhA	1.17	3.70	**0.0006**

β is the covariation factor in a multivariate linear regression model. Here, the covariant is sex. probability valued below 0.05 are bolded.

**Table 7 genes-15-00752-t007:** Association between samples and *TRPV1* H1/H2 haplotypes for BIVA parameters.

	Rsp		Xcsp		Zsp		PhA	
Groups	T Test	*p*	T Test	*p*	T Test	*p*	T Test	*p*
ITA vs. SSA	0.646	0.4604	0.997	**0.0044**	0.659	0.4426	0.999	**0.0012**
M vs. F	1.000	**0.0001**	0.374	0.8938	1.000	**0.0002**	1.000	**0.0000**
ITA_M vs. SSA_M	0.492	0.6922	0.565	0.5795	0.484	0.7039	1.000	**0.0000**
ITA_F vs. SSA_F	0.998	**0.0031**	1.000	**0.0001**	0.998	**0.0030**	0.981	**0.0235**
SSA_M H1	0.853	0.1861	0.607	0.0985	0.427	0.1815	0.967	0.4376
SSA_M H2	0.922	0.5189	0.682	0.4137	0.589	0.5347	0.761	0.0778
SSA_F H1	0.857	0.8276	0.596	0.5582	0.429	0.8245	0.967	0.4499
SSA_F H2	0.665	**0.0418**	0.938	0.3121	0.662	**0.0429**	1.000	**0.0006**

*p* values relative to Student’s *t* test, calculated for each sex and diplotype. R1: aa vs. bb+ab+bc; R2: aa vs. bb+ab+bc. probability valued below 0.05 are bolded.

## Data Availability

The original contributions presented in the study are included in the article/Appendix A; further inquiries can be directed to the corresponding author.

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
