# Peer review of "Association between Variants of the TRPV1 Gene and Body Composition in Sub-Saharan Africans"

_genes, 2024, doi:10.3390/genes15060752_

Round 1

Reviewer 1 Report

Comments and Suggestions for Authors

In this manuscript, the authors present a study of a single locus related to environmental signaling by means of an in-silico approach followed by direct sequencing of two sample sets with different population ancestries. 

There are some sections in the text that are not correctly justified and that can be confusing for readers. 

  • I do not agree with how the aims are exposed in the last paragraph of the Introduction. The authors merge the objectives of the study with some results. Please, rewrite this section. 

  • Figure 1: this figure should be placed in Results’ section. There are some concepts, as FST values, genotyped regions R1 and R2, that have not been previously explained. In the next paragraph, Fst are explained to be estimated with PLINK, but the reader must go to page 6 to understand the genotyped regions.  

  • Capsaicin was an element discussed in Introduction. However, we do not find mention to PROP until Material and Methods, p. 4, when the perception test is explained. The necessity of this test then is not justified, nor the relationship between PROP “phenotype” and body mass and adiposity. Please, provide more context in Introduction.  

  • The Results section starts with “3.1.In silico analysis”. However, here are reported some results regarding the samples experimentally tested ITA and SSA. This is again very confusing.  

Please, provide more details on the criteria used for the recruitment of volunteers. How was the sub-Saharan and Italian ancestry assessed? 

There are some conceptual mistakes. In p. 8, second paragraph. Replace “polymorphic loci” by “polymorphisms”. In Fig. 2 caption, replace “loci rs10491215” by “variant rs10491215”. The text must be extensively revised for the correct use of “loci”/”locus”. Rs identificators denote variants/SNPs/polymorphisms. 

I think that the following statement should be written with more caution “In this research, two highly polymorphic LD blocks (named H1 and H2) were identified in silico in the ORF and the 5'UTR upstream region as targets of a balanced selection process.” 

For me, it is also difficult to directly assume the following statement from the present study results: “SubSaharan Africans recorded higher Zsp and PhA values, indicative of a higher percentage of fat mass (FM%) and a lower extracellular/intracellular water ratio (ECW/ICW). This can be interpreted as an adaptation to long periods of drought and famine”. The results are drawn from concrete individuals, from ITA and SSA groups, from where we do not have information on their lifestyles.  

There are also an issue that should be more strongly specified, basically in Discussion: the low sample size of the two “populations” analyzed, and the presumably heterogeneity in the specific ancestries of those samples (see Table S1). The statements of Discussion are too strong to rely on such small sizes, and they should be treated with more caution. 

Minor points: 

-Page 2, last paragraph: In this study, we addressed some fundamental questions about the role of TRPV1 channels in [recent] human evolution. 

- P. 3, first paragraph: the concept diplotype should be defined (it is actually described in p. 9, first paragraph). 

- P. 3, third paragraph: remove bold font in “The analytical software PLINK version 1.9” 

- P. 3, third paragraph. Provide citation for DNAsp software.  

- Fig. 3: please provide a caption inside the figure itself. It will improve the understanding of color codes. 

- I do not understand this expression (last one from Conclusions): “...the fixation of a variants in non-Africans" 

-There is a typo in Figure s2 (threshold) 

- Some Figures have low quality (Fig S4)

Reviewer 2 Report

Comments and Suggestions for Authors

Gianni et al. submitted a manuscript on TRPV1 gene variants associated with body composition in humans, highlighting that there is a difference for some haplotypes between males and females. More precisely, the haplotypes called H1-b and H2-b were associated with a higher proportion of fat mass and lower extracellular fluid retention in African females, a correlation not observed in males. The conclusions are consistent with the results obtained, even the sample is not very large.

The manuscript is of good quality, although some aspects of the iconographic presentation and tables need to be fixed.

- Many tables present decimal numbers with a comma instead of a period

- In figure 3 a word does not appear to be in English

- There are numerous typos throughout the text that need to be corrected

- The figures have low resolution. The X and Y axis titles and numbering appear to be blurry.

Reviewer 3 Report

Comments and Suggestions for Authors

The authors of Association between variants of the TRPV1 gene and body composition in sub-Saharan Africans wish to draw attention to a theory that is based on sex-driven balancing selection at the TRPV1 gene's non-coding regions, which has adaptive effects on lipid deposition and water balance.

This work is well structured and the idea is good, but some points need to be clarified:

-There are just 46 sub-Saharan and 45 Italian volunteers, making the sample size relatively small. This reduces the findings' statistical power and generalizability.

-The Italian and sub-Saharan populations are the study's main subjects. Incorporating a wider variety of groups may yield a more thorough comprehension of the impacts of TRPV1 gene variations. Why did you consider only them?

-The study primarily focuses on body composition, even though the TRPV1 gene is linked to a number of physiological processes, including energy balance, thermosensation, and nociception. Further investigation into TRPV1's other roles would be helpful. Please mention it in the introduction section.

-Although haplotype blocks H1 and H2 have been identified, and their correlation with body composition has been highlighted, there has been little discussion of the functional consequences or possible biological mechanisms of these particular haplotypes. Please, discuss it briefly.

-Controlling for confounding variables, which may affect body composition and sensory perception, such as age, food, physical activity, and health status, is not mentioned. Please explaine why.

-Strong stabilizing selection signals (MAF close to 0.50, Tajima's D > +4.5) are mentioned in the abstract, however there is not enough background information or justification given to show how important these results are in relation to the larger genetic landscape.

Comments on the Quality of English Language

Moderate editing of English language required

Round 2

Reviewer 3 Report

Comments and Suggestions for Authors

Tha authors performed the corrections suggested.

Comments on the Quality of English Language

Minor editing of English language required